# Evolution of the Shear Band in Cold-Rolling of Strip-Cast Fe-1.3% Si Non-Oriented Silicon Steel

**DOI:** 10.3390/ma14040775

**Published:** 2021-02-06

**Authors:** Yuanxiang Zhang, Yukun Xia, Hao Dun, Yang Wang, Feng Fang, Yu Zhang, Jiecheng Zhang, Qi Chen, Kuangyu Zhai, Raja Devesh Kumar Misra

**Affiliations:** 1State Key Laboratory of Rolling and Automation, Northeastern University, Shenyang 110819, China; zhangyuanxiang@ral.neu.edu.cn (Y.Z.); 1900485@stu.neu.edu.cn (H.D.); wangyang@ral.neu.edu.cn (Y.W.); fangfeng@ral.neu.edu.cn (F.F.); 20183151@stu.neu.edu.cn (Y.Z.); 20183082@stu.neu.edu.cn (J.Z.); 20183100@stu.neu.edu.cn (Q.C.); 20182836@stu.neu.edu.cn (K.Z.); 2Laboratory for Excellence in Advanced Steel Research, Department of Metallurgical, Materials and Biomedical Engineering, University of Texas at El Paso, EL Paso, TX 79968, USA; dmisra2@utep.edu

**Keywords:** shear band, texture, twin-roll strip casting, Cube, non-oriented silicon steel

## Abstract

Cube texture and microstructural evolution of as-cast non-oriented silicon steel (1.3% Si) during cold rolling and annealing were studied. The results showed that the as-cast microstructure with grain size in the range of 100–500 μm had a weak texture. The strong orientation was mainly located at {100} and {110} planes. A significant content of shear-deformed grains oriented with {110}<110> were obtained by cold-rolling, and many regions oriented with Cube texture were distributed in the shear bands. During cold-rolling, the orientation of the shear-deformed microstructure tilted towards the {111}<112> orientation, while the matrix orientation retained {110}<110>. On further cold-rolling, the residual part of {110}<110> experienced shear deformation, forming more shear bands, strengthening the Cube orientation. During annealing, Cube orientation grains nucleated in the shear bands leading to strong Cube texture, and corresponding B_50_ was 1.83T/1.79T.

## 1. Introduction

The non-oriented silicon steels (0.8–2.0% Si) are widely considered for small and medium-size motor cores and used as a soft magnetic material. Their high efficiency is mainly attributed to the high magnetic inductance and low iron loss that requires the increase of favorable texture and reduction of harmful texture [1]. Cube with two <001> directions is the best orientation, while γ texture is harmful [2].

Recent studies have focused on the formation of Cube, Goss, and γ texture and optimizing magnetic properties by rolling and annealing, or by controlling the initial microstructure and texture of slab [3,4,5,6,7,8]. Park [9] studied 2% non-oriented silicon steel, where Cube and Goss grains were found on α and γ shear bands, Cube grains were less than Goss, and the directed nucleation led to recrystallization of silicon steel. The hot-rolled grains can coarsen by normalized annealing and γ texture component decreases, which promotes shear deformation and the development of Cube and Goss textures. However, this approach does not decrease the γ texture component ratio in the annealed texture, and thus the Goss texture is larger than the Cube texture. Cube texture was also obtained in grains with {100} orientation. Cheng et al. [10] demonstrated that {100} texture was furthermore obtained in columnar grains by cold-rolling, and was weakened by hot-rolling with high reduction and recrystallization process. However, the as-cast strip (thickness 1.5–3 mm) could be cold-rolled directly without hot-rolling, which promoted the formation of sharp {100} texture [11,12,13,14,15,16,17]. As indicated in reference [16], during annealing the abnormal growth of grains with {100} texture appeared, and intensive Cube texture and good magnetic properties were achieved after cold-rolling.

The shear bands in oriented grains is related to the formation of Cube orientation. {110}<110> (rotated Goss) texture can be directly obtained in Cube orientation shear bands [18,19], but the rotated Goss texture cannot be obtained by hot-rolling, which means that it cannot be used as a conventional method. Interestingly, in our previous work [20], we found that in Fe-1.3%Si cast strip, the texture obtained by thin strip casting and rolling (TSCR) process had a strong {110} component (including rotating Goss). During cold rolling process, a large number of Cube orientation shear bands were obtained, which significantly increased the strength of Cube texture in the cold-rolled and annealed structure.

In the present study, the occurrence of Cube orientation shear bands in the rotated Goss matrix during cold-rolling of as-cast strip was systematically observed by EBSD, and the orientation distribution of shear bands in the cold-rolled matrix with {111}<110> orientations was compared. Generally, final cold-rolled annealing steel develops Cube orientation. The study confirmed that TSCR process has an obvious determining role on favorable texture. By studying texture evolution in the continuously cast thin strip, favorable texture components, which are of significance for the perspective of high-efficiency and low-cost of non-oriented silicon steel can be obtained.

## 2. Materials and Methods 

Chemical composition of non-oriented silicon steel is shown in Table 1. Strip of thickness 1.5 mm and width 210 mm was first obtained by cast rolling at 1560 °C, and then cold-rolled by a four-high reversing mill (State Key Laboratory of Rolling and Automation, Shenyang, China). The rolling scheme was 1.5 mm-1.2 mm-0.95 mm-0.75 mm-0.65 mm-0.5 mm-0.35 mm, which was about 20% reduction in each pass from 1.5 mm to 0.35 mm. The texture of samples was measured and analyzed using a Zeiss ULTRA55 scanning electron microscope (Carl Zeiss, Jena, Germany) equipped with Oxford Instruments HKL-Channel 5 EBSD system operating at acceleration voltage of 20 kV and a working distance of 16.3 mm. Recrystallized grains were identified based on the criterion consisting of average in-grain misorientation threshold of 2° and surrounding grain boundary misorientation threshold of 15°. Orientation distribution function (ODF) of samples was measured and calculated at different thickness layers with a Bruker D8 Discover X-ray Diffractometer (Bruker, Karlsruhe, Germany) and TexEval software (2.5.8.0, Bruker-AXS, Germany). Typical texture components of non-oriented silicon steel are shown in Figure 1. Magnetic property was tested using the magnetic material tester (MATS-2010M, LINK JOIN, Ningbo, China). Samples of dimensions 30 mm × 100 mm were tested at 1.5 T and 50 Hz for iron loss P_1.5/50_ and at 5000 A/m for magnetic induction B_50_.

## 3. Results and Discussion

### 3.1. Crystal Orientation Analysis of Fe-1.3%Si Strip

Solidified microstructure and macro-texture are shown in Figure 2. It can be seen from Figure 2a that at high superheat condition the grain size was 200–500 μm. High superheat can promote grain boundary migration, resulting in higher frequency of high angle grain boundaries. The cast steel roller with low cooling rate led to a coarse solidified microstructure, which promoted shear deformation during cold-rolling [21]. The macro-texture distribution at S = 3/4 is shown in Figure 2b (the position along the thickness direction can be defined by a parameter S, as S = 2a/d, where a and d are the distance away from the center layer and the thickness of the sheet, respectively, S = 0 corresponds to the center layer and S = 1 to the surface layer). In Figure 2b there are also some strong orientation points in {100} and {110} plane but the magnitude is not high. On the one hand, sub-rapid solidification inhibited the growth of columnar grains. On the other hand, interface energy and temperature gradient can induce the formation of grains with specific orientation. The grains (<001>∥ND) in Fe-1.3% Si cast strip were not dominant during the solidification process, as expected. The columnar-type grains had {110}<223> and {111}<110> orientations. There were few Cube orientation grains, which indicated that sub-rapid solidification had a significant influence on the formation of texture.

There were only a few Cube grains along the longitudinal direction and surface of the cast strip, and a large number of the new Cube grains nucleated during annealing. Previous studies have indicated that many Cube orientation in shear bands were induced by cold-rolling of grains oriented with {110} [16,20]. Thus, we will discuss the evolution of shear bands during the cold-rolling in some detail.

### 3.2. Orientation Analysis and Formation of Shear Bands during Cold-Rolling

Microstructure and texture evolution at S = 3/4 layer during cold-rolling is shown in Figure 3, which shows that after rolling the matrix was mainly comprised of deformed grains and shear bands. Furthermore, higher reduction promoted shear deformation, as shown in Figure 3a,c. Macro-texture study demonstrated uniform distribution and {100} and {110} texture rotated towards α texture and had strong orientation, such as {100}<011>, {113}<110>, {111}<011>, {111}<112> and {110}<110> etc., as shown in Figure 3b,d. 

The theory of plastic deformation of Taylor (Equation (1)) [22] can be appropriately used to analyze the activation of slip system inside coarse grains while considering relatively low grain boundary energy:
(1)$$M=\frac{\sigma_{x}}{\tau }=\frac{d\gamma }{d\varepsilon_{x}}$$
where *M* is Taylor factor, $\sigma_{x}$ is normal stress, τ is shear stress, γ is shear strain, $\varepsilon_{x}$ is normal strain. 

Taylor factor of grains with {100}<011> was the smallest, and dislocation slip occured relatively easily. The {110}<110> grains had the largest value (*M* value 4.25), and simultaneously the relatively high value was found in grains with {111}<223>, {110}<111>, {111}<110>, and {111}<112> (and so on). This suggested that the grains were resistant to deformation, and thereby needed more deformation energy. 

The orientation intensity of Cube grains did not decrease with increase in reduction, but the general trend was a gradual increase. This behavior is related to grain reorientation with geometric softening of {110}<110>~{110}<223> orientated grains with high value of Taylor factor during deformation.

Generally, tensile stress decreases with increase of deformation because of grain rotation, and is referred to as grain reorientation with geometric softening [23]. Work hardening, geometrical hardening, and grain reorientation with geometric softening all may take place during plastic deformation. Dislocation pile-up and activation of slip system is known as work hardening. Whereas, the geometrical hardening is related to grain rotation, which is beneficial for further deformation. This occurred preferentially in grains with low M value. When many slip systems are activated, the grains with high M values tend to deform via local shear deformation and the M value decreases. Nguyen-Minh [18] calculated the starting condition of grain reorientation with geometric softening of 110}<110> orientation grains in Fe-1.2% Si, and Cube orientation shear band was observed. The {110}<110> orientation grains in hot-rolled sheet are a kind of special grain with relatively low volume fraction and they have little effect on microstructure and texture evolution. However, in strip casting the sub-rapid solidification enhances the component ratio of {110}<110>~{110}<111> with a high M value, and leads to evolution of Cube microstructure during rolling. 

Humphreys [24] believed that microstructure of shear bands is complicated such that the evolution of shear bands cannot be described in detail. He even considered it as an amorphous plastic deformation behavior. In reality it is difficult to characterize shear bands with large deformation as cellular structure with high dislocation density, and the difference between a single crystal system and polycrystalline system is large. The {110}<110> orientation grains are stable, experience shear during intensive shear deformation, and the grains produced by strip casting are coarse. The evolution of shear bands observed during rolling is shown in Figure 4 and Figure 5. 

The main slip plane of {110}<110> grains parallel to the direction of applied stress rendered a high M value, which meets the necessary condition of grain reorientation with geometric softening. In addition, there are many plastic instability conditions in a polycrystalline system, which makes shear deformation a preferred plastic deformation. As shown in Figure 4a, “Shear band 1” and “Shear band 2” crossed {110}<110> and {111}<112> matrix grains. But this kind of shear microstructure was not observed at grain boundaries including {110}<110>, {001}<110>, {110}<223>, which suggested that grains tend to rotate and activate a new slip system because of similar orientation between adjacent grains and limited dislocation slip. The shear deformation can traverse grain boundaries. The shear band with low M value orientation, such as Cube, Goss, and {210}<001> orientations, had a 25–40° tilt angle with RD, as shown in Figure 4b. The results demonstrated that the local severe deformation is the driving force behind low-elastic-modulus <001> orientation of cellular structure of shear bands, which rotated along the deformation direction of matrix for maintaining continuity of the microstructure. Shear deformation is related to partial high strain rate deformation. Based on minimum strain energy principle, restriction occurs from the two edges of the matrix, which can influence orientation of micrometer region during intensive deformation process [25]. The grain reorientation with geometric softening after shear deformation, as shown in Figure 4c, indicates that the M value of shear microstructure is apparently lower than shear bands on two edges.

Grain orientation after shear deformation exhibits strong stability, as shown in Figure 4d,e. It is generally acknowledged [26] that {110}<110> orientation with weak stability will change to {111}<110> when reduction is greater than 30%. This rotation tendency is obvious when shear deformation occurs. Furthermore, the deformation of strip was inhomogeneous because large grain size makes soft orientation grains with low M value require some work hardening to reach critical value so that shear can occur through grain reorientation with geometric softening. 

Figure 5 shows the microstructure and orientation distribution of as-cast strip after 67% cold-rolled reduction. It can be seen that there were a few narrow new shear bands with various orientations marked with black arrows in the matrix and they had an angle of 25°−40° with the RD direction. This suggested that {110}<110> orientation matrix with high M value and some shear bands continued to deform by grain reorientation with geometric softening and produced a new shear microstructure with low M value, as shown in Figure 5a. Secondary new shear microstructure can go through developed shear zone and {110}<110> grains in some cases, as “shear band 3”. Cube and some other adjacent orientations tilted towards {311}<136>, then {112}<110>~{111}<110> stabilize, as shown in Figure 5b–d.

When the reduction was increased to 77% in Figure 6, shear bands were gradually broadened, the orientation gradually changed from hybrid orientation to unified Cube orientation, and the shear band density was increased. The angle with RD direction was in the range of 0°−25° which is smaller than the range of 25°–40° for new shear bands, as shown in Figure 6a,b. Actually, the Cube orientation microstructure of new shear bands in {110}<110> grains was not inherited, because it could easily rotate towards other stable orientation. Therefore, Cube orientation shear microstructure formed in the last stage of deformation had a significant effect on nucleation and growth of Cube grains during annealing. As is shown in Figure 6c,d, the fraction of Cube orientation was increased. This phenomenon is consistent with the Nguyen-Minh’s experimental results [18], indicating that shear deformation easily occurred inside the matrix grains with high M value orientation, and shear bands are oriented with low M value orientation. With further deformation, the slip system in the low M value orientation grains is activated, then the grains changed from “soft orientation” to “hard orientation” because of dislocation pile-up and multiplication. They tend to deform by shear deformation in the next step to ensure continuity of plastic deformation. 

The microstructure of shear bands in {111}<110> and {111}<112> deformed grains are shown in Figure 7 and Figure 8. In the conventional process, {110}<110> grains are difficult to inherit, while the {111}<110> and {111}<112> grains can be retained. This influences recrystallized texture [7,9,27]. In the solidified microstructure, the fraction of {111}<110> and {111}<112> orientation grains were low, but they were high in the cold-rolled sheet. This phenomenon indicates that the texture fraction is controlled by cold-rolling. Below 77% reduction, the shear microstructure in {111}<110> grains was not developed, and a large number of traces of dislocation slip were observed by EBSD, as shown in Figure 7a.

Some cellular structure with approximate {110}<001> orientation inclined by 25°–30° off RD. Moreover, the cellular structure appeared in interaction area of shear bands and initial boundary, as shown in Figure 7b. The reason why the Cube texture exists in traditional hot-rolled process is that grain boundaries between fine grains have a strong effect on Cube cellular structure. As is shown in Figure 7c,d, it is the {111}<110> orientation that is intensive, which reveals the stability of this texture.

The above discussion suggests microstructural evolution of characteristic orientation of α texture and indicates that formation of {111} <112> texture results from the activation of slip systems along {112} plane (parallel to RD). For example, Goss grains rotated 38.9° around TD axis to <110> axis would acquire {111}<112>. The orientation of shear bands in {111}<112> grains below 77% reduction is shown in Figure 8a,b. The density of sharp Goss orientation shear bands with 15°–20° angle with RD was greater than in {111}<110> and {110}<110> deformed grains. The developed shear bands included some Cube orientation units, while the orientation of new shear bands was Goss and other neighboring orientations in Figure 8c,d. A high intensity of Goss orientation suggested that there is a specific relationship between {111}<112> grains and Goss grains, which can be attributed to the recrystallization process in oriented silicon steel [28].

### 3.3. Microstructure, Texture and Magnetic Property of Annealed Sheets

The microstructure and macrotexture of the annealed sheet processed by cold-rolling and followed by annealing are shown in Figure 9 and Figure 10, respectively. The average grain size was 60 μm after annealing at 900 °C for 5 min, as shown in Figure 9a. Based on ODF, intensity of Cube orientation was highest along the thickness direction while Goss and {111}<112> orientation were less, as shown Figure 9b. 

Further analysis of texture suggested that the intensity of Cube orientation was the highest, followed by Goss and {111} <112> was the lowest, as shown in Figure 10a–c and Table 2. Orientation distribution has an obvious effect on the homogeneity of magnetic properties and the difference between magnetic induction value of ND and TD was less than 2.3%, as shown in Table 3. Compared with commercial W470 with P_15/50_ along the RD and TD was 4.0 W/kg and 4.3 W/kg, respectively. B_50_ of RD was 1.70 T and TD was 1.68 T, and the magnetic properties were improved. In Dou’s work [29], the best P_15/50_ of RD/TD was 4.5/4.6 W/kg and corresponding B_50_ along RD/TD was 1.78/1.76 T, and in An’s work [30], the best P_15/50_ along RD/TD was 4.235/4.467 W/kg and corresponding B_50_ along RD/TD was 1.798/1.743 T. By comparison, the magnetic properties in our study are superior by tuning the microstructure and favorable texture.

## 4. Conclusions

We have discussed the effect of shear deformation on recrystallized microstructure, texture and magnetic properties of Fe-1.3% Si strip during cold-rolling. The primary conclusions are as follows: (1)Solidified microstructure was coarse and uniform in the as-cast strip. The {100} and {110} texture and some approximate Cube texture were present, which did not influence the distribution of Cube orientation grains in annealed sheets produced by cold-rolling.(2)In the strip, some {110} orientation grains had high Taylor value, such that more deformation energy obtained by shear deformation was necessary to induce plastic deformation. When the reduction was 67%, the shear bands in {110}<110> grains were developed. When reduction was below 77%, the shear bands in {110}<110> and {111}<112> grains were completely developed.(3)The underlying reason for grains with high Taylor value to shear and acquire η texture such as Cube and Goss orientation is attributed to grain reorientation with geometric softening. Shear bands in {110}<110> grains mainly consisted of Cube orientation, while it was Goss orientation in {111}<112> grains. Shear band formation in {111}<110> grains was less.(4)With increase in percentage reduction, strong Cube orientation occurred in {110}<110> grains that experienced shear deformation and the major orientation of shear bands was Cube. Further cold-rolling led to rotation of grains and promoted shear of {110} <110> deformed grains.(5)Cube and Goss orientations were intensive in S = 0 and S = 1/2 layers, especially Cube. The annealed sheets had excellent magnetic properties.

## Figures and Tables

**Figure 1 materials-14-00775-f001:**
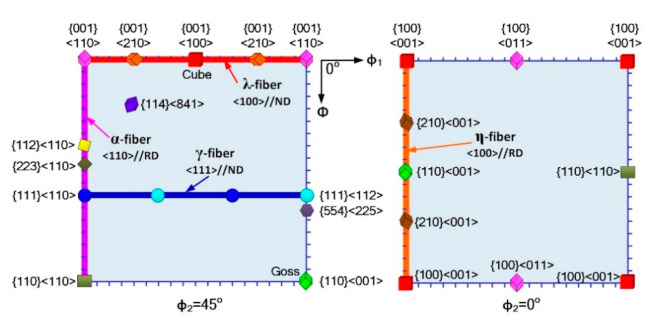
Typical texture components of non-oriented silicon steel in constant φ2 = 0° and φ2 = 45° ODF sections.

**Figure 2 materials-14-00775-f002:**
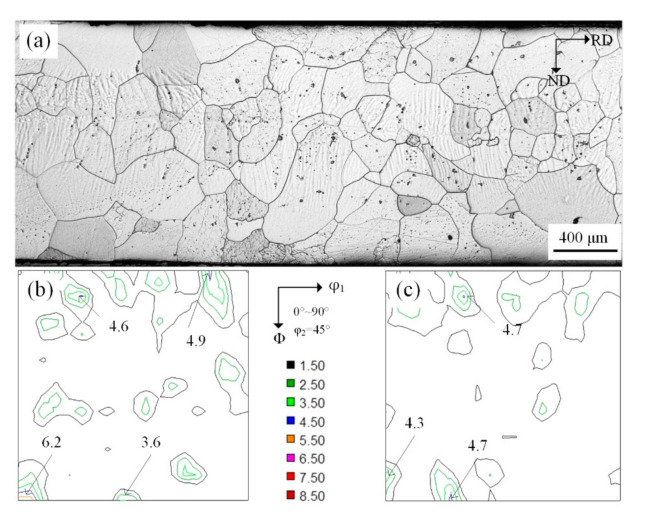
Microstructure (**a**) and constant φ_2_ = 45° sections of ODF at S = 7/8 (**b**) and S = 1/2 (**c**) of the Fe-1.3% Si cast strip.

**Figure 3 materials-14-00775-f003:**
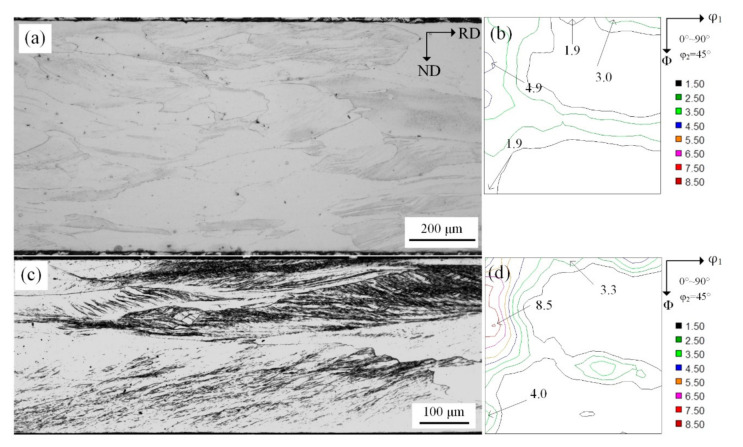
Microstructure of 50% (**a**) and 77% (**c**) reduction and constant φ_2_ = 45° sections of ODFs at subsurface layer (S = 7/8) of 0.75 mm (**b**) and 0.35 mm (**d**) cold-rolled sheets respectively.

**Figure 4 materials-14-00775-f004:**
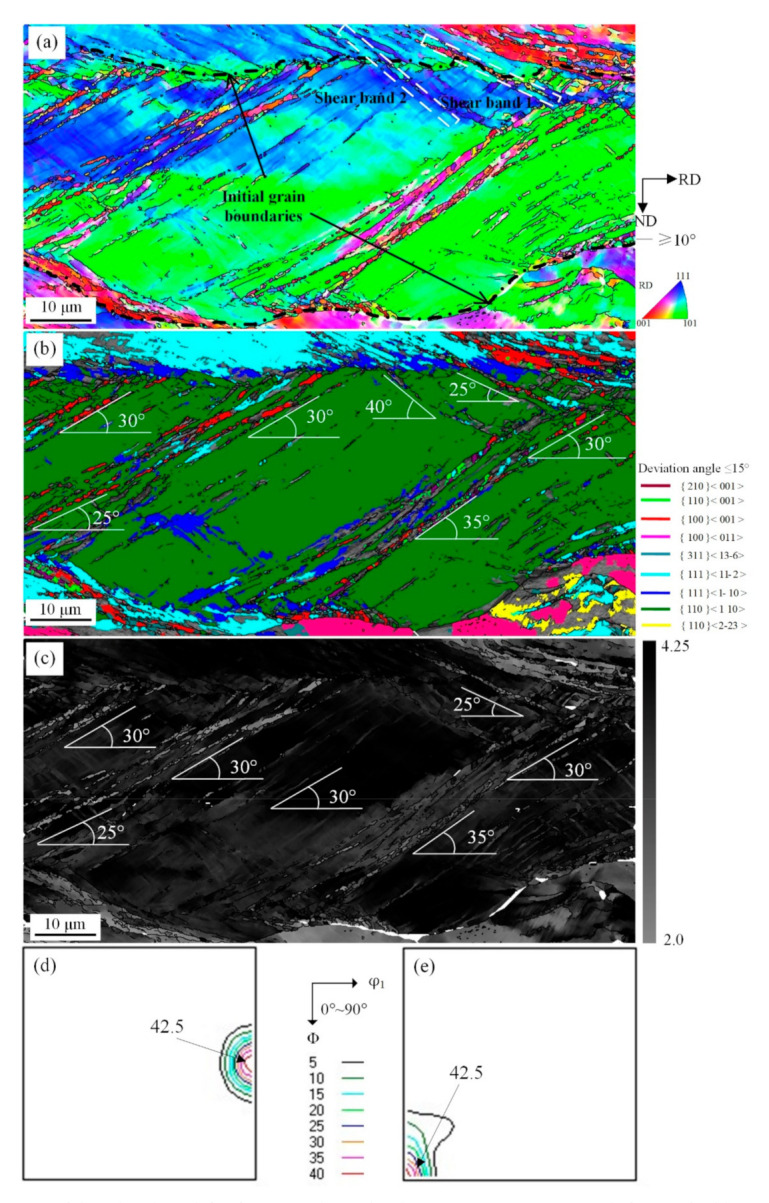
EBSD orientation maps of shear bands with {110}<110> under 57% reduction (**a**) ESBD inverse pole figure of cold-rolled sheet; (**b**) several texture components colored in orientation maps of cold-rolled sheet (deviation angle ≤ 15°); (**c**) Taylor factors of shear bands; (**d**) ODF of EBSD data presented at φ_2_ = 0° section; (**e**) ODF of EBSD data presented at φ_2_ = 45° section.

**Figure 5 materials-14-00775-f005:**
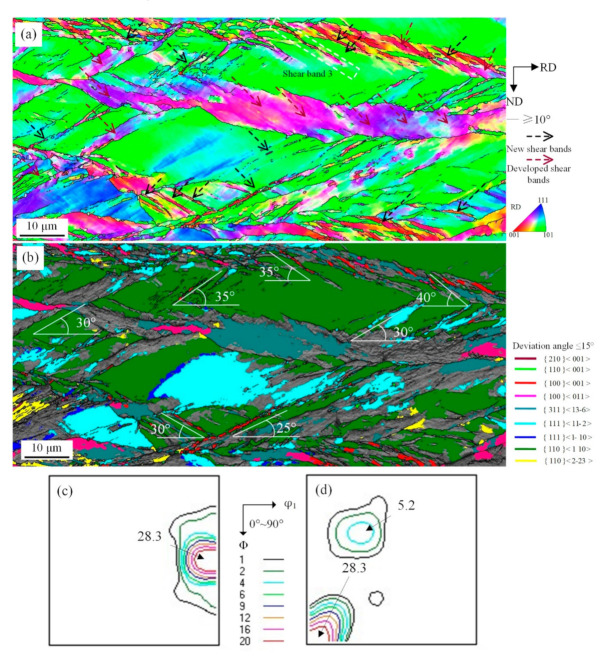
EBSD orientation maps of shear bands with {110}<110> under 67% reduction (**a**) ESBD inverse pole figure of cold-rolled sheet; (**b**) several texture components colored in orientation maps of cold-rolled sheet (deviation angle ≤ 15°); (**c**) ODF of EBSD data presented at φ_2_ = 0° section; (**d**) ODF of EBSD data presented at φ_2_ = 45° section.

**Figure 6 materials-14-00775-f006:**
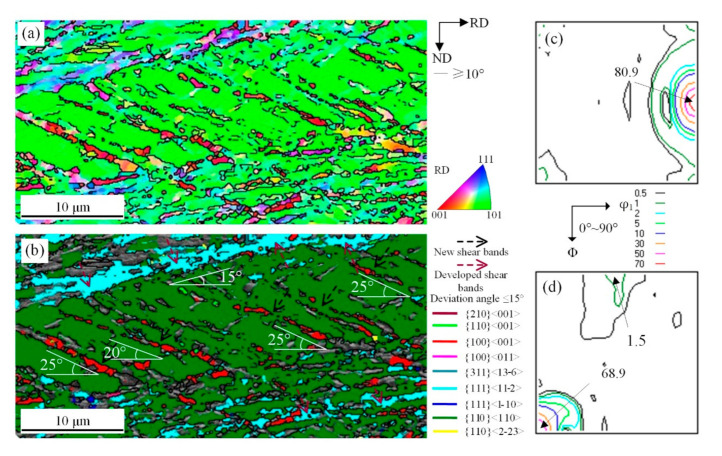
EBSD orientation maps of shear bands with {110}<110> under 77% reduction (**a**) ESBD inverse pole figure of cold-rolled sheet; (**b**) several texture components colored in orientation maps of cold-rolled sheet (deviation angle ≤ 15°); (**c**) ODF of EBSD data presented at φ_2_ = 0° section; (**d**) ODF of EBSD data presented at φ_2_ = 45° section.

**Figure 7 materials-14-00775-f007:**
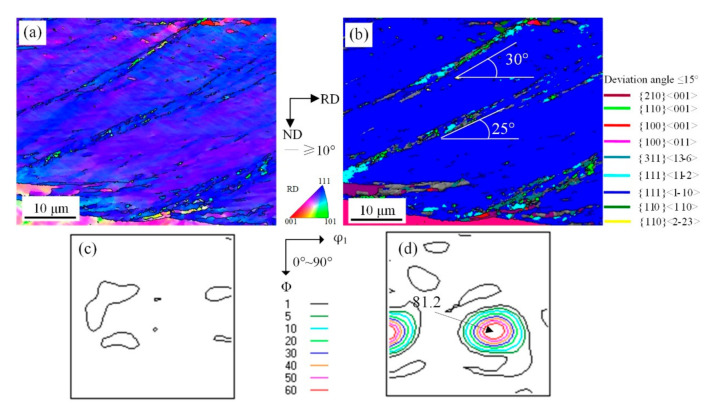
EBSD orientation maps of shear bands with {111}<110> under 77 reduction (**a**) ESBD inverse pole figure of cold-rolled sheet; (**b**) several texture components colored in orientation maps of cold-rolled sheet (deviation angle ≤ 15°); (**c**) ODF of EBSD data presented at φ_2_ = 0° section; (**d**) ODF of EBSD data presented at φ_2_ = 45° section.

**Figure 8 materials-14-00775-f008:**
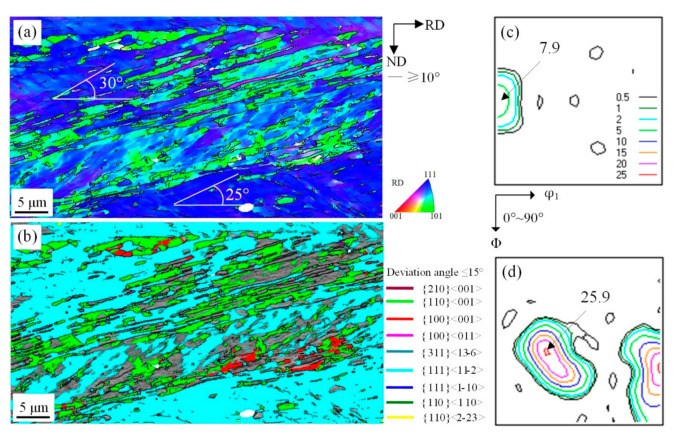
EBSD orientation maps of shear bands with {111}<112> under 77 reduction (**a**) ESBD inverse pole figure of cold-rolled sheet; (**b**) several texture components colored in orientation maps of cold-rolled sheet(deviation angle ≤ 15°); (**c**) ODF of EBSD data presented at φ_2_ = 0° section; (**d**) ODF of EBSD data presented at φ_2_ = 45° section.

**Figure 9 materials-14-00775-f009:**
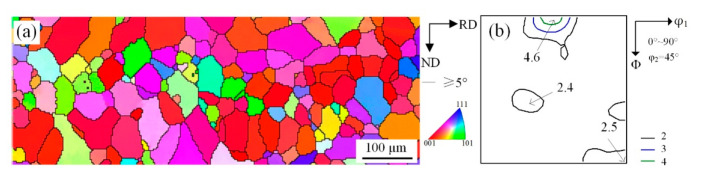
EBSD IPF map (**a**) and φ_2_ = 45° section of ODF (calculated from left IPF map) (**b**) of 0.35 mm annealed sheet.

**Figure 10 materials-14-00775-f010:**
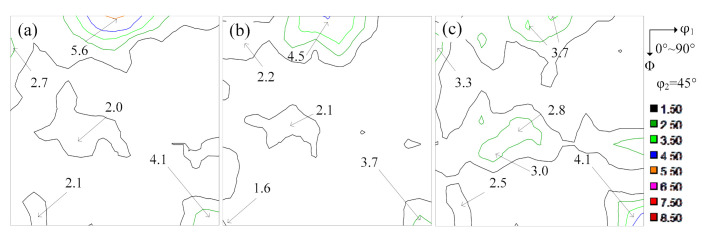
The annealing macro-texture of 0.35 mm strip-cast non-oriented silicon steel: (**a**) S = 0; (**b**) S = 3/4; (**c**) S = 1/2.

**Table 1 materials-14-00775-t001:** Composition of non-oriented silicon steel (mass fraction %).

Fe	C	Si	Mn	S	Al	N	O
Bal.	0.0036	1.3	0.31	0.0034	0.22	<0.004	<0.003

**Table 2 materials-14-00775-t002:** The volume and ODF value of characteristic annealing texture (deviation ≤ 15°) of strip-cast non-oriented silicon steel: (**a**) S = 0; (**b**) S = 3/4; (**c**) S = 1/2.

Characteristic Texture	(a) S = 0		(b) S = 3/4		(c) S = 1/2	
	%Volume	ODF	%Volume	ODF	%Volume	ODF
Cube	8.120	5.640	7.491	4.530	4.678	2.990
Goss	4.746	3.390	3.952	2.590	6.090	5.200
{111} < 112>	2.595	2.140	2.579	1.180	4.008	2.920

**Table 3 materials-14-00775-t003:** Magnetic properties of 0.35 mm strip-cast annealing specimen. Adapted with permission from ref. [20]. 2021 Elsevier.

P_15/50_/W·kg^−1^		B_50_/T	
RD	TD	RD	TD
4.1	4.3	1.83	1.79

## Data Availability

The data presented in this study are available on request from the corresponding author. The data are not publicly available due to technical or time limitations.

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
