# Peer review of "Evolution of the Shear Band in Cold-Rolling of Strip-Cast Fe-1.3% Si Non-Oriented Silicon Steel"

_materials, 2021, doi:10.3390/ma14040775_

Round 1
Reviewer 1 Report
In the paper entitled “Evolution of the Shear Band in Cold Rolling of Strip-cast Fe-1.3%Si Non-oriented Silicon Steel”, the Authors present the results of studies which concerns describing texture and microstructure evolutions of the as-cast non-oriented silicon steel (1.3% Si) during the cold rolling and annealing. The title and abstract are appropriate for the content of the text. Furthermore, the article is well constructed, the experiments and analysis were well performed. In my opinion, the manuscript gives adequate contribution in the investigated field and it can be accepted for publications after minor revision. However, I have a few comments which may be allowed to improve the quality of your work. The details are attached below.
1) Line 51 - it should be Philips
2) Authors should provide more details about the texture analysis (e.g. measurement parameters, software name, whether correction for defocusation effect and background intensity were applied)
3) Histograms should be added to the given grain size information
4) In this case quantitative texture analysis would be valuable information.
Reviewer 2 Report
Dear authors,
I think that the readership will surely find the results of general interest.
From my perspective, the manuscript requires a review of a number of sentences. Personally I found it hard to read at multiple points. In the end it is probably relevant for you to get the message across and unambiguously.
Many authors refer to the Taylor factor with the letter M, and you do so as well. But in my opinion this is not done clearly at the outset, Eq. 1. I propose to use the expression, Taylor factor, consistently, or simply refer to it as M, to improve the presentation.
On lines 53 and 54, the authors probably mean that the magnetic flux strength is of 5000 A/m.
The conclusion referring to geometrical softening as the cause of shear of grains with high Taylor factor seems somewhat ambiguous to me. You appear to state that the high density of dislocations is directly tied up to the geometrical softening, although the story in the rest of the article bears a more intricate story. I may have misunderstood the argument, of course.
Due to the introduction I expected a deeper discussion about the magnetic properties. Additionally, only data from the 0.35 mm thick sample is presented, but there is also a 0.5 mm thick sample. You may like to comment about the reason to only present one. Or, perhaps it is only one sample cold-rolled in two stages down to 0.5 and then to 0.35 mm? It was not clear to me.
Good luck with the publishing process.
Best regards,
Reviewer 3 Report
The work is devoted to the study of the evolution of the structure and shear bands in silicon steel. The work is of scientific interest and practical interest. However, the authors do not fully understand what texture is. The work constantly talks about grain texture, but such a concept does not exist. Authors should understand the concepts of grain orientation and material texture. The work should be revised based on the comments written above, and after that its full consideration is possible.
Round 2
Reviewer 3 Report
The authors have significantly revised the manuscript.
Authors should comment on the following issues before the article is published: 1. The authors claim that after annealing, the sheets contain up to 10% of grains with a favorable orientation. The authors need to comment on how many oriented grains, in their opinion, need to be obtained to achieve the best magnetic properties. 2. The authors should compare the obtained magnetic properties with the results of other works to understand their uniqueness
